# Understanding the Survival Ability of Franchise Industries during the COVID-19 Crisis in Malaysia

**Nurul Ashykin Abd Aziz** [1,*], **Mohd Hizam-Hanafiah** [2], **Hasif Rafidee Hasbollah** [3], **Zuraimi Abdul Aziz** [1] and **Nik Syuhailah Nik Hussin** [1]

1   Faculty of Entrepreneurship and Business, Universiti Malaysia Kelantan, Locked Bag 36, Pengkalan Chepa, Kota Bahru 16100, Kelantan, Malaysia; zuraimi@umk.edu.my (Z.A.A.); niksyuhailah@umk.edu.my (N.S.N.H.)
2   Faculty of Economics and Management, Universiti Kebangsaan Malaysia, Bangi 43600, Selangor, Malaysia; mhhh@ukm.edu.my
3   Faculty of Hospitality, Tourism and Wellness, Universiti Malaysia Kelantan, Locked Bag 36, Pengkalan Chepa, Kota Bahru 16100, Kelantan, Malaysia; rafidee@umk.edu.my
*   Correspondence: ashykin.a@umk.edu.my

**Abstract:** Since the world was hit by the COVID-19 pandemic crisis that began in December 2019, many industries have been affected, including the franchise industry in Malaysia. Thus, the COVID-19 pandemic has had a great impact on business survival. Direct effects can be seen in reduced income, job losses, changes in customer preferences, and business relationships between franchisors and franchisees. Some franchises have had to close their operations, and others still struggled to survive during the pandemic crisis. In addressing this situation, the role of government is crucial in supporting the resilience of these franchisor entrepreneurs in an increasingly worrisome situation around the world. However, the existing literature that focuses on the role of government in developing countries such as Malaysia is still poorly understood. In addition, a study of the Malaysian franchising industry during the pandemic crisis is still inadequate, especially concerning the government's role in the survival of local franchises during the pandemic era. Therefore, understanding the role of the government in advocating the survival of local Malaysian franchises is worth studying. A qualitative research approach was applied through multiple cases involving twelve (12) franchise business owners and four (4) franchise-related agencies in Malaysia. In-depth interviews were conducted in exploring this topic. Thematic analysis has been used by applying "Atlas.ti" in analysing the data. Hence, the findings have indicated four themes from the grounded data. There are: (i) financial assistance; (ii) virtual franchise exhibition; (iii) training and support; and (iv) business development grants. This study is expected to highlight the role of government as well as agencies involved with the franchising industry in improving policies, strategies, and programs to ensure the viability of the franchise industry during periods of pandemic outbreaks.

**Keywords:** COVID-19; financial crisis; franchise industries; survival ability

## 1. Introduction

The COVID-19 virus has shaken the world for the past two years. The pandemic broke out in China in December 2019 and spread across the world since March 2020. As of 8 February 2022, 225 countries were affected, 398,093,224 people had been infected, with 5,768,797 deaths, and 317,637,345 patients recovered [1]. In addition to the serious implications for people's health around the world, this pandemic has had a huge impact on businesses and the economy. According to Alves et al. [2], small businesses are generally vulnerable to the crisis, yet there is still much that is not known about how their businesses survived throughout the pandemic crisis of COVID-19. Furthermore, the ongoing pandemic led to the bankruptcy of many well-known brands in many industries as economic transactions between countries were put on hold. According to the information shown in Figure 1 released by the Malaysian Insolvency Department (MDI) in December 2020,

overall Malaysian bankruptcy data was reported at 711,000 units in December, up from 419,000 units in November. The information is classified as part of the Malaysian Global Database (Impact COVID-19) [3].

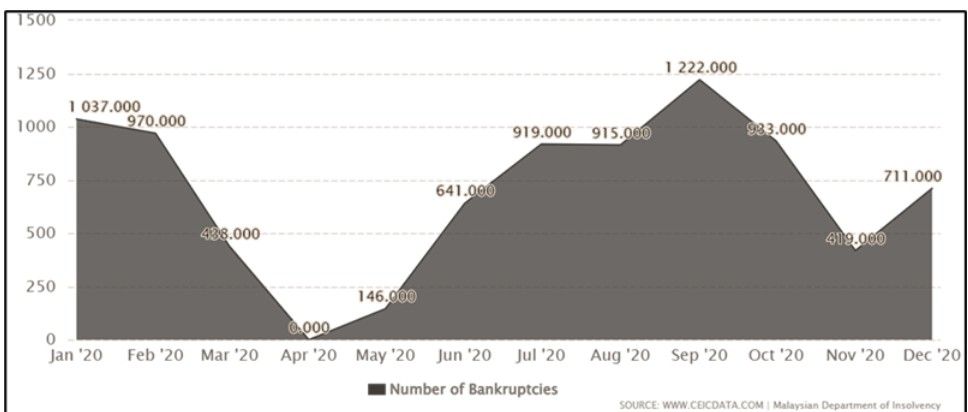

**Figure 1.** Number of Bankruptcies. Source: Malaysian Department of Insolvency.

There were no more customers, as all residents were instructed to remain at home during the lockdown [4]. The pandemic has harmed most countries' economies, but perhaps has been even more critical to emerging economies. Although Malaysia's economy appears to be heterogeneous, it has some general characteristics as well, such as weak institutions and legal provisions, lower economic development levels, and higher levels of financial and social risk [5,6].

For the franchising industry, the COVID-19 outbreak also dealt a huge blow to the survivability of franchise businesses. As a commercial and social model, franchising has a variety of economic and social benefits, including employment creation, economic development, and the growth of entrepreneurship [7,8]. Additionally, the immediate effects on income, employment, and social goals are most visible in growing and developing markets [7,9,10]. Furthermore, Bretas and Alon [11] claimed that the COVID-19 pandemic had an effect on political as well as societal ramifications throughout the world. The franchising industry, which is mostly comprised of retail and service enterprises, is an example of one that has been severely affected. The experiences of franchising stakeholders show the franchising model's capabilities in such scenarios.

According to the International Franchise Association [12], roughly 32,700 franchised businesses in Malaysia have shut down in the first six months following the COVID-19 outbreak as of August 2020. A total of 21,834 enterprises were partially shuttered, with 10,875 remaining closed permanently. In every twenty (20) small firms there will be one (1) firm that permanently closes down for the following six (6) months. Hence, it is predicted that without major government advocacy, another 36,000 franchised units will not be able to survive. While most franchise businesses are related to services and retail, they cover a wide spectrum of business activities and are affected to different degrees by the crisis [6,13]. As suggested by Fabeil, Pzim, and Langgta [14], further analysis is needed with other respondents in future studies to fully understand the survival mechanisms adopted by businesses in response to the pandemic crisis. Furthermore, a better understanding of how pandemics affect entrepreneurship and entrepreneurs ability to cope with various pressures is needed. Due to the increasing emphasis on the knowledge economy, the role of government institutions is important, and this needs to be explored further. Future studies could develop a more inclusive support model to demonstrate the role of government in supporting schemed geared toward business survival [15,16]. Meanwhile, Boiral, Brotherton, Rivaud Leo, and Guillaumie [17] emphasized the uncertainty about the pandemic, and its implications and the likelihood that it will ensue. Hence, it is important for organizations to plan measures that can be adopted in the event of another pandemic outbreak in the future. Therefore, the role of organizations needs to be explored more widely in helping businesses

to survive. Thus, an understanding of the role, factors, and effects of pandemic on the survival of Malaysian home-grown franchises are vital to research. Therefore, this research is guided by a broad research question: "How do franchise-related agencies play their roles in the survival of the Malaysian home-grown franchises during a pandemic crisis?"

## 2. Literature Review

### 2.1. The Development of Franchising in Malaysia

According to George [18], franchising is the fastest method to grow and expand a business, whereby this is the easiest way to do a business. In addition, franchising means growing and expanding one's own businesses by letting other people invest and hence, it is the best method to cut short the learning curve. In Malaysia, franchising began in the early 1940s with Bata and Singer, and then was followed by automobiles and petrol station dealers. Nevertheless, the development of franchising proceeded faster when fast food service restaurants such as A&W (since 1963), Kentucky Fried Chicken (since 1970), and McDonalds (since 1981) opened their business operations rapidly in Malaysia. The development of Malaysian franchises began in the early 1980s with a focus on the food and beverage sectors (e.g., Sate Ria, Marrybrown), petrol stations (e.g., Petronas), the automotive sector(e.g., EON) and craft companies (e.g., Royal Selangor). To support the development of Malaysian franchises, the government appointed Majlis Amanah Rakyat (MARA) to facilitate the activities and tasks related to these efforts [19].

Furthermore, in supporting the development of Malaysian franchising, the Franchise and Vendor Division was authorized on 8 May 1999 under the Ministry of Entrepreneur Development (MED), previously known as the Ministry of Public Enterprises. In addition, under this division, there wase a Franchise Unit and a Vendor Unit. Moreover, the Franchise Unit had managed the Franchise Development Programme, which was transferred from the Implementation Coordination Unit of the Prime Minister's Department, while the Vendor Unit managed the Vendor Development Programme under the Ministry of International Trade and Industry [20].

According to the records of the Franchise Development Division [20], these two programmes were then transferred to establish the Ministry of Entrepreneur and Cooperative Development in order to meet governmental policy requirements and to accelerate the accomplishment of goals in creating Bumiputera Commercial and the industrial community. Later, when the Ministry of Entrepreneurism and Cooperative Development was dissolved in 2008, the Franchise Development Division's role was transferred to the Ministry of Domestic Trade, Co-operatives, and Consumerism (MDTCC), which is now known as the Ministry of Domestic Trade and Consumer Affairs (MDTCA).

### 2.2. Institutional Theory

This study adopted institutional theory in order to get a fundamental understanding of the research question. In this theory, Di Maggio and Powell [21] highlighted the role and function of government agencies and trade associations, both of which are vital to industry definition, and the emergence of networks, mutually formal and informal, which support policies regarding the industry. Barthélemy [22] explained that institutional theory highlighted the differences in the usages of franchises where to some extent, while institutional judgment also simplifies the relationship between agency considerations and franchise decisions. Clearly, through educational programmes, franchisors offered systematic education for prospective master franchisees and potential franchisees. In addition, through financial assistance for franchises from financial institutions, it enabled the easy entry of potential franchisees. Without doubt, it explained the importance of the government's function in helping franchising businesses that expands into the international market [23,24].

### 2.3. Survival of Franchise Business

As the franchise has grown to become a well-known and prominent business platform throughout the world, it has piqued the interest of scholars and policymakers. The



decision of entrepreneurs to engage in franchise businesses improves their chances of survival in a saturated market [25–28]. The primary goal of entrepreneurship is to generate jobs and improve the economy [29]. According to Michael and Combs [30], it is vital to have a better understanding of the factors influencing franchise survivability in order to encourage satisfaction in franchise relationships. Furthermore, Kosonova and Francine Lafontaine [31] asserted that the contract term of a chain may impact the growth and survival of franchise chain, and that certain attributes may be influenced by other variables. Bordonaba-Juste, Lucia-Palacios, and Yolanda Polo-Redondo [32] also highlighted that these additional variables may affect franchise survival by relying on the features of the franchisee's experience.

In general, the COVID-19 pandemic has a direct influence on the economy, particularly the franchise industry's sustainability. The franchising business will not be the same after the pandemic, even if the business reopens. While the immediate repercussions might have passed, the long-term economic consequences will continue to fluctuate for years [11,13,33,34]. Some franchising brands may go out of business, while others may be severely impacted and might have to shut several of its franchisees. On the other hand, despite the COVID-19 pandemic that continues to spread over the world, numerous brands are prospering. In the context of this pandemic, for franchisors and franchisees need to work together, in addition to receiving support from certain parties to continue to survive [34].

Meanwhile, Bretas and Alon [11] highlighted that franchising businesses in Brazil proved the franchise model's strength in a COVID-19 pandemic situation. This is based on significant data from webinars with businesses in the food services, education, retail, and business-to-business service sectors in Brazil, as well as reports from commercial and franchise entities. Scholars have explained how the COVID-19 pandemic has impacted the franchising industry. They explain the processes performed, the talks between the supplier and the landlord, the business model modification, the influence on franchisor-franchisee relationships, and long-term survival [11,35].

Moreover, the COVID-19 pandemic has affected the business climate and commercial management practices around the world because it is unpredictable [36]. The pandemic issue directly resulted in the closure of the hotel and tourism sectors. This has resulted in a company failure rate that has increased exponentially. The franchise hospitality business now has a huge influence and challenges with the basic concepts of knowledge management implementation in facing the COVID-19 pandemic to survive. Strategic knowledge management implementation strategies may not only lead the particular franchise, but also transform an organisation's performance and competitiveness for long -term sustainability [36].

## 3. Research Methodology

### 3.1. Study Design

In identifying cases and specific types of case studies to be implemented in a study, researchers should consider whether to conduct a single or multiple case studies in order to gain an in-depth understanding of the phenomena that will be studied. Additionally, context is also another matter to be considered [37–40]. Furthermore, Baxter and Jack (2008) described that researchers study various cases to understand the differences and similarities between cases. When making comparisons between cases, researchers can also provide literature of important influences from the differences and similarities for these cases [41]. Vannoni [41] argued that the evidence made from various case studies needs to be and reliable. Moreover, by studying various case studies, researchers can build a more convincing theory when the proposal is based on empirical evidence.

Obviously, many cases provide the opportunity to explore research questions and the evolution of a broader theory, as claimed by Eisenhardt [42]. For collective or multiple case studies, data collected should be sufficient and flexible to enable each case to be described in detail. This needs to be done by the researcher before considering the similarities and differences that arise in cross-case comparisons. It is vital that data sources from

different cases, as much as possible, are comparable, even though they may differ in some ways [43]. The present study adapted a multiple case study approach, as these create a more convincing theory when the suggestions are more deeply grounded in empirical evidence. As mentioned before, this study's main aim is to identify how the government plays a role in the survival of a Malaysian home-grown franchise during a pandemic. Therefore, the use of qualitative methodology is appropriate for the researcher in to achieving the objectives.

### 3.2. Setting and Participants

Fundamentally, the sampling frame in a qualitative study is a combination of convenience and purposive and theoretical sampling. According to Dornyei [44], convenience sampling is a category of nonprobability sampling where the target population meets a certain criteria, namely: it is easy to access, available at a given time, or a person is willing to participate in the study. Purposive sampling is a technique used in qualitative research to identify and select information-rich cases for the most effective use of limited resources [45]. In line with this, multiple cases consisting of thirteen (13) franchising companies and three (3) government agencies related to franchise development in Malaysia were selected for the study. The researcher interviewed the prominent top management of these franchising companies and agencies as individuals who could provide the required information such as the role of government, incentives and support given to franchise entrepreneurs during a pandemic crisis.

Individuals, groups, or organisations would be the unit of analysis for this research. The unit of analysis is determined at the preliminary phase in the study since the conceptual framework, data collecting procedures, and sample size are all dependent on it [46–48]. In this research, the franchisors and top management from the franchise development-related agencies are the key informants as well as decision makers, and are also actively engaging in the franchise industry. Charmaz [49] highlighted that a very small sample could still produce an important study, and the factors influencing this study could include a quality interview and an in-depth analysis. The following factors could also affect the sample size: (i) the researchers' experience and expertise in interviews and subject areas are important components in reducing sample size and in achieving saturation [50,51]; and (ii) the appropriate selection of participants [52,53]. Boddy [54] argued that qualitative research is often associated with the development of a deeper understanding rather than a broad one, especially when conducted under a non-positivist paradigm, as it involves in-depth psychology or a constructivist approach to the research.

Meanwhile, the sample size is sufficient with the strength of information, depending on the purpose of the study, the specificity of the sample, the sustainable use of theory, the quality of dialogue, and the analysis of strategies [55]. Furthermore, Vasileiou, Barnett, Thorpe and Young [56] have claimed that the provision of sample size in qualitative research is limited and does not depend on the number of interviews. Defensive sample sizes are most often supported by references to saturation principles and pragmatic considerations [57,58]. The data of saturation can be accomplished with at least six (6) interviews and small samples, depending on the size of the population sample in the qualitative study [59–61]. For this study, the saturation data was achieved in the 12th interview with franchising business owners and then supported by four (4) interviews with agencies related to franchise development in Malaysia.

This study also used triangulation techniques, where the interviews are made from two sources, namely from franchise companies and agencies related to franchise development, as well as data from printed sources such as annual reports and franchise acts. In qualitative research, triangulation refers to the use of various methodologies or data sources to build a thorough knowledge of phenomena. Triangulation is seen as a qualitative research approach for testing the validity by bringing together data from disparate sources [62,63]. Moreover, triangulation can enhance research by providing a range of datasets in explaining the various elements of an interest topic [64,65].

### 3.3. Data Collection

For the present study, the primary data collection method is the interview. According to Cassell [66], even though the interview technique can vary widely based on its applicability and structure, this method is widely used in organisational research. Data for this study were gathered through sixteen (16) in-depth interview sessions (details in the supplemental materials). All in-depth interviews were done individually with franchising business owners and franchise development organisations in Malaysia. These interviews took place between September 2020 and October of 2021. A group of researchers guided by the research protocols had performed each in-depth interview session. With the respondents' agreement, audio was captured for later research purposes. In-depth interviewing is a qualitative research approach that entails conducting intense individual interviews with a limited number of respondents in investigating their viewpoints on a certain topic, programme, or issue [67,68]. The goal of these interviews are to learn about how people see, comprehend, and make sense of their lives, as well as how they assign meaning to certain experiences, events, and issues [69].

The lead researcher guided the interview discussion with the respondents for the data collecting method, while other member assistants recorded interviews and jotted down field memoranda, providing information for subsequent research. Each session began with a broad question designed to elicit debate between the researcher and the respondent. "What are the roles of franchise-related agencies in assisting the survival of the franchise business during the pandemic period?" is an example of such a question. The main researcher complimented the respondents and provided tokens to them re for their involvement at the end of the interview session, and conclusions were reported based on the interview findings. As a result, protocol talks were held with franchising business owners and franchise-related agencies for this study, and each interview and discussion lasted from 50 to 60 min.

### 3.4. Data Analysis

For the present study, thematic analysis was adopted to analyse the data gathered. Thematic analysis is defined as a method that allows researchers to identify and organise relevant themes and subthemes that can later be used as units of analysis [70,71]. Next, the researchers reread the datasets is to familiarize themselves with the data, and then they explore the meanings associated with the concepts that emerged from the statements of participants in the study [72]. Thematic analysis is one of the types of methods in qualitative data analysis for research methods that have taken place in various fields [73]. According to Labra, Castro, Wright, and Chamblas [74], thematic analysis offers flexibility, but a rigorous approach to subjective experience as a method of promoting social justice and combating inequality is ideal for research in social work.

The hematic analysis done in this study involves six phases [74]. The first step begins with the task of transcribing the audio recordings of the individual or group interviews conducted in a study. The next step involves the researcher continuing the initial reading of the transcript to search for the most significant parts of the participant's testimony of the phenomenon being studied. At the level of analysis available, the researcher will use the information identified as relevant in the first step to generate the initial code. Initially, researchers began to collect data elements according to perceived equations or patterns and this was labeled as the initial code [74]. Next, the theme or category is derived from data elements or sequences of words that can serve as synoptic and precise representations of the comments of the participants interviewed with regard to a phenomenon being studied. As such, the theme consists of coded data grouped together according to equations or patterns [48,75].

In the fourth step, a comprehensive description of the phenomena studied requires a systematic review theme to be identified in the third step. However, for discussion purposes, the fourth step is identified as different, and after the third phase, in practice the researchers are already familiar with thematic analysis and will regularly perform

the analysis of two phases simultaneously. The fifth step consists of two main stages. At the first level, themes and subthemes undergo a review. The thematic matrix must be re-analyzed thoroughly to assess the validity of the hierarchical relationship and confirm whether the terms given at both levels are accurate with the meaning indicated by the code. It is important to check that a name is given to the themes consistently, so that there is no ambiguity about their accuracy. The second stage is interpretive and consists of the definition of concepts and subthemes to be applied to a deep analysis in the sixth step. In the sixth step, presentations and discussions that differ from each other are presented [74].

The data analysis was carried out in accordance with these six theme phases. The researchers assessed the data by closely reviewing the transcripts of the interviews word by word. The researchers then re-read the transcript, extracting all themes, and grouped them into main groups with smaller subthemes based on the equation. Finally, each theme was evaluated and interpreted considering the study's setting. Themes and subthemes were chosen with the confirmation of all researchers, validated by the relevant literature, and then presented to respondents for data validation.

## 4. Finding

The research population included 12 (twelve) franchising business owners from three business categories, as well as four (4) franchise-related agencies in Malaysia. Table 1 provides a summary of the sociodemographic characteristics of the participants in this study.

**Table 1.** Sociodemographics of Participants.

| Participant Code | Category | Role | Participation |
|---|---|---|---|
| P1 | Education and learning centre | Franchisor | In-depth interview |
| P2 | Education and learning centre | Franchisor | In-depth interview |
| P3 | Education and learning centre | Franchisor | In-depth interview |
| P4 | Food and beverages | Franchisor | In-depth interview |
| P5 | Food and beverages | Franchisor | In-depth interview |
| P6 | Food and beverages | Franchisor | In-depth interview |
| P7 | Food and beverages | Franchisor | In-depth interview |
| P8 | Food and beverages | Franchisor | In-depth interview |
| P9 | Food and beverages | Franchisor | In-depth interview |
| P10 | Food and beverages | Franchisor | In-depth interview |
| P11 | Self-service Laundry | Franchisor | In-depth interview |
| P12 | Self-service Laundry | Franchisor | In-depth interview |
| P13 | Franchise-related agency | Top Management | In-depth interview |
| P14 | Franchise-related agency | Top Management | In-depth interview |
| P15 | Franchise-related agency | Top Management | In-depth interview |
| P16 | Franchise-related agency | Top Management | In-depth interview |

Based on the results of the interviews, the findings show that there are four themes that emerged from the grounded data to meet the objectives of this study. Table 2 shows the distribution of themes.

### 4.1. Theme 1: Financial Assistance

The beginning of the discussion session was based on the participants' experience with the types of responsibilities that agencies associated to the franchising industry play in order to support the majority of the franchising firms affected by the pandemic attack since the end of 2019. According to the participants, various incentives and stimulus

packages have been implemented by the government to ensure that entrepreneurs affected during the COVID-19 pandemic can survive and continue their businesses. In addition, the franchise-related agencies are committed and concerned to ensuring the viability of business in the country by providing comprehensive business support assistance in meeting the challenges posed by measures to curb the spread of COVID-19. This is aligned with these quotes:

> "Franchise-related agencies are very important in helping franchise entrepreneurs by providing various financial incentives . . . these agencies offer financial loans to support our business capital, as well as financial assistance to stay survive in the competitive industry."—P12

> "We agree that many good roles and efforts have been shown by franchise related agencies in Malaysia . . . we are very grateful for their efforts including providing financial assistance . . . to maintain in this franchise industry is not easy and financial support that given like a loan . . . this really motivates us."—P6

**Table 2.** Distribution of Themes.

| Themes | List of Participants | | | | | | | | | | | | | | | |
|---|---|---|---|---|---|---|---|---|---|---|---|---|---|---|---|---|
| | **P1** | **P2** | **P3** | **P4** | **P5** | **P6** | **P7** | **P8** | **P9** | **P10** | **P11** | **P12** | **P13** | **P14** | **P15** | **P16** |
| Financial Assistance | | | | | | / | | | | | | / | | / | / | |
| Virtual Franchise Exhibition | / | | | | | | | | / | / | | | | / | / | / |
| Training and Support | | / | | / | | | / | / | | | | | / | | | / |
| Business Development Grant | | | / | | / | | | | | | / | | / | | / | |

/: This symbol (/) indicates we marked as "tick" and has no other meaning.

Furthermore, the business assistance provided also includes assistance to all entrepreneurs including the franchise industry in managing their cash flow. Among the initiatives introduced were loan moratoriums, salary subsidies and employment incentives designed to maintain the capacity of employees and companies, loan assistance grants and financing guarantees. Importantly, the assistance provided by the government in aid packages and economic stimulus can provide relief to the cash flow pressures faced by traders.

> "As a related agency, the development and growth of the franchise industry is also part of our organizational goals . . . during this pandemic season, many businesses are affected and financially unstable . . . what we can offer is financial assistance such as loans . . . this moratorium . . . is to reduce the burden and promote their survival."—P14

> "For us, business survival for franchise businesses is very important when tested by a pandemic . . . many are affected and affected . . . what we offer is intensive financial assistance to encourage them to stabilize the business while reducing the financial burden borne . . . they can apply for a franchise financing scheme . . . "—P15

### 4.2. Theme 2: Virtual Franchise Exhibition

The Franchise International Malaysia (FIM) exhibition, which has been held in Malaysia every year since 1994, is a worthwhile effort. It has offered several advantages to the franchising firms because most Malaysian franchisors attended this international show to exhibit their products and services while also hunting for possible business partners. Aside from that, the participants reported that most of them attended international franchise

exhibitions in other countries as a worldwide business platform to develop networking with bigger groups of possible investors or partners that have little or no understanding about the franchise brands. For instance:

> "When the movement control order is implemented, our physical movement is limited ... cannot congregate ... so we greatly appreciate the efforts of these government agencies in organizing virtual franchise exhibitions ... even virtually, we can use this platform to look for potential franchisees and new investors."—P1

> "This virtual franchise exhibition is very helpful in connecting us with other countries even virtually ... on this platform, we can spread information about our brand, find potential franchisees and establish relationships and business networking without borders ... "—P10

> "Participation in this virtual franchise exhibition means a lot to us ... although we can't physically interact, but this platform gives us the best results in finding potential franchisees and investors ... "—P9

Based on the participants, organizing a virtual exhibition can save a lot of money on travel, accommodation, promotional items, and other costly expenses typically incurred by franchise businesses. Due to the constraints of the COVID-19 pandemic, various bans, controls, and directives that had been issued by the government resulted in limitations to physical movement and travel. Thus, the organization of virtual exhibitions contributes to cost reduction. This also allows more franchising business owners to participate in the virtual franchise fair, which increases their chances of furthering their business expansion opportunities and finding potential investors. This is in line with this quote:

> "We started organizing this virtual franchise exhibition in 2020 ... this is one of our initiatives to support the involvement of various franchise brands either locally or internationally ... this virtual exhibition is cost effective, we are compliant ... in addition we provide a platform for franchise entrepreneurs even in the constraints of the COVID-19 pandemic."—P16

> "The organization of this virtual franchise exhibition benefits the Malaysian franchise industry ... although there are physical movement constraints, this platform is seen to be cost -effective and effective ... franchisors and prospective franchisees who participate in this platform do not have to spend a lot of cost ... they can only communicate online ... "—P14

> "Our agency is also involved in this virtual Malaysian international franchise exhibition ... for us, this is a good effort in helping to boost the franchise industry even in a pandemic situation ... we ourselves offer consulting services if there is a franchisor or this prospective franchisee knows more about the industry."—P15

*4.3. Theme 3: Training and Support*

Based on the discussions with the respondents, all of them agreed that most franchising entrepreneurs were highly affected by the COVID-19 pandemic. Due to the outbreak, the government had to implement strict movement control orders and almost all non-essential business sectors were ordered to close. Alternatively, some franchisees, especially in the service sector, such as education and learning centres, have had to switch to online mode. However, it is not as easy as one might think due to their preparation. Thus, among the efforts of franchise-related agencies to support these franchise operators is by organising online training and providing ongoing support.

> "During the movement control order, our physical movement is limited ... if we want to attend or organize training, there are constraints ... so we really hope if there is a virtual program or training ... or maybe a forum or webinar organized by the- certain parties can help us to be more motivated and continue our business during this pandemic period ... "—P2

> "COVID-19 came suddenly and caused most sectors to close . . . we were also impressed . . . alternatively, we had to use an online platform . . . through ongoing training and support from relevant agencies, we got a lot of information on how to run the business online."—P8

As franchising businesses have been severely affected by the COVID-19 pandemic, the role of relevant agencies in focusing on entrepreneurial development and strategies in business survival were very vital. Therefore, the training process has been highlighted as a way for franchisees to plan how to best conduct their business operations under the constraints of movement order control and limited permitted business operating hours. Additionally, a good training program organised by these franchise-related agencies is seen from a positive angle where it will help develop the franchise businesses affected during this pandemic so that the franchise runs better, and the business owners improve their capabilities and identify the next training steps needed. Most franchise-related agencies agree that this training program is particularly important for franchisees who want to improve their outlet management skills and knowledge during this prolonged pandemic season. Here are the related quotes:

> "For now, and since the pandemic crisis, we offer a lot of training, webinar sessions and support online . . . it can be said that quite often we organize webinar sharing sessions . . . we invite industry experts to share how to run a business and what strategy to deal with this situation."—P13

> "On our part, we do our best to help and provide continuous support to the Malaysian franchise sector . . . through cooperation with various other franchise-related agencies, we strive to conduct virtual training . . . with this we can provide input to these franchise entrepreneurs."—P16

Moreover, part of the ongoing training and supports provided by franchise-related agencies to these franchise businesses is operational management training. Essentially, management training in a franchise business describes the formal and ongoing efforts undertaken by franchise-related agencies to train franchise operators in improving their business performance during pandemic situations through various business methods and programs. In managing a franchise business during this pandemic, entrepreneurs need to have highly specialised management skills for long-term professional development. This is aligned with this quote:

> "When we are suddenly hit by this pandemic situation, we only realize that there are things that we have enough skills to handle the operation . . . therefore, training and ongoing support is important . . . for us, it is important to follow any form of training, in order to improve our skills and knowledge."—P4

> "For us, the ongoing support provided by the franchise-related agencies is important . . . through involvement in training, we gain new knowledge, create business networking and subsequently gain new skills that may add value to us."—P7

### 4.4. Theme 4: Business Development Grant

The last theme discovered from the grounded data based in this study involves the business grant. According to the participants, a business grant is money given to a franchise business from a franchise-related agency. It is certainly an attractive financial consideration for those with little cash available for start-ups, growth, or expansion. There are grants available for all types of franchise business owners, whether franchisors, franchisees, or the master franchise. Typically, grants are available when a government agency or non-profit agency chooses to help a franchise business start or continue its business. A grant may be provided to encourage minority entrepreneurship in the community. Another can be provided to fund research and development in a particular industry, to reward innovation,

or to promote a franchise business. This is one of the government's initiatives in helping franchise entrepreneurs affected by the pandemic.

> "We are offering several incentives including a grant scheme . . . these grants support business continuity among franchise businesses."—P15

> "Offering grants to any franchise business is actually an advantage to them . . . this is because they can use the funds provided to develop and ensure their survival ability especially in this pandemic situation . . . "—P13

The participants have explained that financing business operating costs during a pandemic is difficult, and in fact is impossible for most franchisees, as the majority will face issues such as the lack of customers, limited business hours and even strict standard operating constraints imposed by the government. Business grants are seen as one of the initiatives of franchise-related agencies that offer the opportunity to raise seed capital without the financial stress of loans having to be repaid. The business grants offered a variety of advantages to franchising businesses, but these vary depending on the scheme. This financing grant is very helpful especially when the world is hit by a pandemic and they serve to help increase the survivability of the franchise business (e.g., micro and affordable franchise development programmes). In addition, business owners will gain confidence when they know that their ventures have been publicly endorsed by the scheme operator and use the funds to continue business operations, promotion, and business continuity.

> "For us, these franchise-related agencies are very helpful to us . . . not only provide financial incentives, but there are also agencies that offer business development grants . . . this is quite helpful and motivates us to continue and grow our business."—P3

> "Funding through this business development grant is very good . . . for any franchise entrepreneur who is just starting out, this is a great opportunity to increase survival ability in a competitive market."—P7

> "Business development grants are a good effort . . . credit should be given to these agencies who work hard to help the development of Malaysia's franchise industry."—P11

## 5. Discussion

The aim of this study is to explore and understand the role of institutions and agencies related to franchise development in helping franchising businesses affected by the COVID-19 pandemic. This study was based on the experiences of the institutions and franchisees that were involved in the main challenges for most businesses, including the survival ability of the franchise sector.

According to the respondents, the most important challenge for business survival during the pandemic outbreak is to maintain cash flow. As such, most participants agreed that franchise-related agencies are very helpful in providing financial assistance such as loans and moratoriums. This finding is supported by a previous study by Ismail and Othman [76] on 118 respondents from small and medium enterprises (SMEs) that were surveyed using a structured questionnaire as an instrument. Findings from this study showed that government support programs had a positive effect on business growth among SMEs in Melaka. This research provides insights into SME support agencies in increasing the effectiveness of relevant government support programmes such as financial assistance. Another study by Rungani and Potgieter [77] highlighted that SMEs success is positively and significantly connected with financial support from either the public or private sectors. While financial assistance is critical, it must be reorganised to include practical features.

Furthermore, the private and public sectors should work together to establish an environment that supports the efficient use of finance, which will enhance firm performance. A previous study conducted by Assefa [15] revealed that 44 percent of small enterprises will collapse during the first month of closure restrictions during a COVID-19 pandemic.

Furthermore, just six (6) percent of enterprises have enough cash to survive for a year. In this context, the report recommends that the government should refrain from implementing complete closure measures in the absence of government assistance initiatives. As a result, the study looked at the relevance of COVID-19-based special loans, payment suspensions, limited fund withdrawals, and tax exemptions and penalty payments. The COVID-19-based special loan is the most important government support scheme, followed by the suspension of interest and principal payments.

Next, respondents stressed that the franchise development related agencies were very helpful to them during the pandemic by organising virtual franchise exhibitions. This finding is supported by past studies done by Abou-Shouk, Zoair, Farrag, and Hewedi [78], who claimed that that venue design, facilities, staff, available information and comfort had a positive impact on exhibitors. Furthermore, Pu, Xiao, and Du [79] claimed that price promotions will have a beneficial impact on sales performance, information collecting, and company image construction, and promotions are not exhibitor pricing. Furthermore, the conclusions of these researchers' findings revealed that price exhibitor marketing does not have a beneficial influence on the creation of customer connections, however no price promotion does.

Meanwhile, Pinandita, Nofrizaldi, and Shabiriani [80] discovered that the exhibition is one of the relatively major activities that necessitates direct engagement with a large number of people. The exhibition, which drew a large crowd, could not be held while the outbreak was still underway. It is unclear when the pandemic will cease, but more innovative approaches to present an exhibition are required. Virtual technology is the way out of the traditional showroom's new physical reality constraints. During the epidemic, the virtual exhibition is projected to have the opportunity to sustain the creative process.

The respondents also claimed that franchise development-related institutions greatly assist franchisees by conducting online training and providing ongoing support. This finding is aligned with a past study by Adeiza, Bo, Abdul Malek, Ismail and Mohd Harif [81]. They found that training and support management services have a substantial impact on the success of the franchisees' businesses at both the early and growth phases. However, their research also demonstrated that training has the greatest influence early on, but this benefit fades as the franchisee learns the intricacies of the business. Management of the service, on the other hand, ensures that it remains relevant at all levels of the franchise firm. Thus, Bui, Jambulingam, Amin, and Nguyen [33] supported that franchisor support plays a very important role in simplifying the relationship between entrepreneurial orientation, market orientation and franchisee performance in the COVID-19 pandemic situation. Moreover, franchisor support services may include assistance with site selection, assistance with hiring, the provision of initial and ongoing training, financing support, management services, operations management services, marketing and promotion support, and research and development support [82–84].

On the other hand, the respondents have indicated that franchise related agencies are very important in playing their role to help the franchise industry during the pandemic by offering business development grants, and this is very helpful to the franchisors and franchisees involved. According to a past study by Quintiliani [85], the author discovered empirical evidence that showed a strong relationship between government grants and the diversity of strategies with the growth of enterprise value. Additionally, Srhoj, Lapinski, and Walde [86] revealed that business development grants had a large positive influence on capital shares, bank loans, intermediate inputs, and value added on average, but had no indication of a beneficial effect on productivity metrics, sales, employment, average wages, or inventories. Yet, major outcomes are contingent on the positive impact of small businesses. Furthermore, there are positive effects from winning the business plan competition on a company's capital stock, employment, sales, and profits in previous studies examining the impact of business development grants [87,88].

As highlighted by Srhoj et al. [88], business development grants have favourable impacts on company sales and employment, as well as survival likelihood and access to

bank loans, but only for one-year-old enterprises; no benefits were seen for two-year-old firms. Both studies showed that business development grants are appropriate for small businesses. For instance, research and development (R&D) grants have largely beneficial effects on the company's R&D expenditure, innovation and performance [89,90]. Anther past study by Srhoj et al. [88] indicated that grant plans have a favourable effect, which is especially important for small businesses. Estimates of this dose response function have also revealed that the percentage of total grants for business income must be large enough for the grant to be effective. An initial calculation shows that the benefits outweigh the direct program expenses. Moreover, Srhoj et al. [86] claimed that capital-constrained enterprises that can increase their output if their capital or labour constraints are eased. Grant support is given as one of the reasons why firm performance metrics are projected to improve with the lifting of capital limitations. Receiving a grant, on the other hand, verifies a firm's quality to banks and makes it simpler to receive a loan. Based on these findings and discussions, this study proposes several propositions (Table 3).

**Table 3.** Propositions.

| Themes | Propositions |
|---|---|
| Financial Assistance | P1: Consistent financial assistance by government is positively associated with the survival ability of the Malaysian franchising business community during the pandemic. |
| Virtual Franchise Exhibition | P2: The better the international exhibitions organized to help Malaysian franchising business, the higher the Malaysian franchising propensity to survive during the pandemic situation. |
| Training and Support | P3: Training and support activities are positively associated with the survivability of Malaysian franchising businesses during the pandemic. |
| Business Development Grant | P4: The greater the access to business development grants for Malaysian franchising businesses, the higher their chances of survival during the pandemic. |

## 6. Conclusions

In brief, this study has highlighted the roles played by franchise-related institutions in helping to strengthen the country's franchising industry, especially during the pandemic era. It is clear that these franchise-related agencies advocate for franchising entrepreneurs in continuing and sustaining their businesses. The findings indicated four roles of agencies related to franchise development, which are: (i) financial assistance; (ii) virtual franchise exhibition; (iii) training and support; and (iv) business development grants.

Franchising entrepreneurs received financial assistance and guidance from government agencies as well as from banking and financial institutions. Many business industries were affected by the movement control orders enforced by the government and this resulted in many businesses, including franchises, not continuing their operations. As a result of this, there are franchising entrepreneurs who had encountered financial problems and are burdened with costs to bear, besides not generating incomes similar to those that they did before the pandemic. Thus, most of the franchising businesses depend on financial assistance and incentives provided by the agencies involved. In addition, the effort to organize a virtual franchise exhibition is seen as a very effective effort in assisting franchise entrepreneurs in continuing their business activities during the pandemic. This virtual exhibition is seen as an international platform and can bring together many leading franchising brands despite the constraints of physical meetings.

Physical movement is very limited during a pandemic due to the implementation of movement control orders by government, and this greatly complicates the conduct of training for businesses. However, with many initiatives by various agencies related to franchise development, this constraint was addressed by organizing online training such as webinars. In addition, ongoing support is also provided so that these franchising entrepreneurs can maintain their momentum despite being tested during the pandemic

outbreak. Finally, the provision of a business development grant is also one of the essential roles of franchise-related agencies. For franchising entrepreneurs affected by the pandemic, funding assistance in the form of development grants is perceived as an important mechanism to support their businesses and to help it survive. Moreover, business development grants are also imperative in supporting the growth and expansion of franchising businesses in both the domestic and global markets.

## 7. Academic and Practical Implications

There are academic and practical implications based on the findings from this study. As for the academic implications, the findings have indicated themes that have enriched the existing theories related to the role of institutions in advocating the survival of the franchise industry as well as becoming a reference in exploring a more robust role of development-related agencies for the survivability of franchisors and franchisees in Malaysia. As for practical implications, the results of this study can be used as a guide for the franchise development agencies in creating more initiatives in boosting and empowering the franchise industry during any pandemic situation. Furthermore, this study has a practical implication for the franchise industry players in increasing their continuation of franchise networking in maintaining the survivability of franchise development in Malaysia.

## 8. Limitations and Future Research Directions

The findings of this study may not be generalisable for areas other than franchisors and government support agencies. This is due to the research that only focuses on the findings during the COVID-19 pandemic. It is suggested that future studies that explore the survival ability of franchisees or master franchisees in different situations be carried out in order to do a comparison of outcomes based on specific situations. This future study is recommended to be done by using other methodologies such as a grounded theory approach or a phenomenology approach for a more diverse and in-depth exploration of themes as well as through the use of quantitative methods. Thus, it can help researchers to discover and test the relationships of franchise agencies in Malaysia.

**Supplementary Materials:** The following supporting information can be downloaded at: https://www.mdpi.com/article/10.3390/su14063212/s1, Part A: General Questions; Part B: Roles of franchise-related agencies.

**Author Contributions:** N.A.A.A. and H.R.H. focused on writing—original draft preparation, methodology, software and analysis. M.H.-H. and Z.A.A. prepared conceptualization and resources. N.S.N.H. and Z.A.A. conducted validation and data accuration. N.A.A.A. and N.S.N.H. managed project administration. M.H.-H. and H.R.H. conducted writing—review and editing. All authors have read and agreed to the published version of the manuscript.

**Funding:** The publication of this paper is financing by UMK-Fund 2020 (STRG) Grant fund (R/FUND/A0100/01231a/001/2020/00842) from Universiti Malaysia Kelantan, led by Nurul Ashykin Abd Aziz, along with other grant members, namely Nik Syuhailah Nik Hussin, Hasif Rafidee Hasbollah, Zuraimi Abdul Aziz and Mohd Hizam Hanafiah (Universiti Kebangsaan Malaysia).

**Institutional Review Board Statement:** Not applicable.

**Informed Consent Statement:** Not applicable.

**Data Availability Statement:** Not applicable.

**Acknowledgments:** We would like to express our appreciation to Universiti Malaysia Kelantan for the UMK-Fund (STRG) research grant provided. Also, many thanks to all the authors for their efforts and support in the preparation of this article. Hopefully this article will benefit all parties. particularly the franchise industry in Malaysia and all franchise players.

**Conflicts of Interest:** The authors declare that they have no conflict of interest.

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
