# Peer review of "Understanding the Survival Ability of Franchise Industries during the COVID-19 Crisis in Malaysia"

_sustainability, doi:10.3390/su14063212_

Round 1
Reviewer 1 Report
The article has no raised hypotheses.
Article sounds scientifically grounded in language, but the findings are such that could be expected even without research and already are applied more than a year by companies: like virtual exhibitions or training.
So, the appropriate construct not always guarantee the value of results. Due to research results depending on research preliminary hypotheses the scientific value of the article becomes from average to low.
Author Response
Dear Prof/Assoc.Prof/Dr,
First and foremost, thank you for reviewing our manuscript. Also, thank you for the constructive comments provided to further solidify the writing of our study results. Here we have attached the responses for each comment. Thank you for your attention and time. We really appreciate it.

Reviewer 2 Report
This manuscript is clear and relevant to the field, except for the issues I have raised. Franchise business relations are not commonly researched on, so the paper will fill the gap in this area. It is scientifically sound and appropriate to test hypotheses.
1. The authors must consider overhauling the entire paper's English language. The following are some of the few sentences with poor grammar:
- In Line 28, the word "that" is missing to give good meaning.
- In line 76, ''another 36,000 franchised units may be not survived''.
- In line 183, ''researchers should consider whether to conduct one case studies''
- Line 282, identifying and organize relevant and relevant themes and subthemes
2. The results are reproducible, the authors may need to attach the sample interview questionnaires in the appendix.
3. In properly showing the data, the authors might consider providing more analysis from the software package of Atlas. ti .
4. As for the conclusions consistent with the evidence and arguments presented, the authors must consider including academic and practical implications of the study.
Note: This study is not like the usual straight forward research studies, it is an innovative paper on Franchise, which is rarely written about. Also, the Atlas.ti software (https://atlasti.com/) is a computer program used mostly, but not exclusively, in qualitative research or qualitative data analysis.
Author Response

(The authors gave the same response as above.)

Reviewer 3 Report
This research addresses how government and franchise-related agencies should play their role to make a franchise industry survive during a pandemic crisis. I think it is a very timely and interesting research. Nevertheless, some modifications are needed before publication.
First, author(s) should not make a conclusion that all franchise companies have been experiencing business crisis. Some franchise companies (e.g., Buger King in East Asian area) have rather increased their profits during pandemic period. So, author(s) needs to revise some sentences such as “This Covid-19 pandemic crisis has a negative impact on the business survival.”
Second, despite some meaningful implications of this research, there seems to be more limitations that should be rectified in the future research. So author(s) needs to make more effort to present better ideas for future research.
For instance, author(s) mentioned that “Direct effects can be seen in reduced income, job losses, changes in customer preferences, and business relationships between franchisors and franchisees.” Here, I think the business relationships between franchisors and franchisees are very important to survive a pandemic crisis. However, present research does not consider this issue in investigating the role of government and agencies, so it should be mentioned as a limitation. Recent research argues that pursuing a balanced profitability between franchisors and franchisees is critical for the sustainability in franchising (Lee et al., 2021). It also shows how the marketing decision on the number of store and advertising expense affects the franchisor and franchisee’s performance, respectively. To overcome a pandemic crisis, government and agencies may suggest a policy to make two counterparts (franchisor and franchisee) pursue a balanced profitability. Author(s) should enrich a part of conclusion by referencing to many recent research.
Lee, E.; Kim, J.-H.; Rhee, C.S. Effects of Marketing Decisions on Brand Equity and Franchise Performance. Sustainability 2021, 13, 3391. https://doi.org/10.3390/su13063391
That’s all. Great job on your research. I hope you get a good result finally.
Author Response
Dear Prof/Assoc.Prof/Dr,
First and foremost, thank you for reviewing our manuscript. Also, thank you for the constructive comments provided to further solidify the writing of our study results. Here we have attached the responses for each comment. Thank you for your attention and time. We really appreciate it.

This manuscript is a resubmission of an earlier submission. The following is a list of the peer review reports and author responses from that submission.